mathematical modelling/health and disease and epidemiology

COVID-19, spatial diffusion, mechanistic-statistical model, non-local transmission, immunity rate

**Author for correspondence:**
L. Roques
e-mail: lionel.roques@inrae.fr

# A parsimonious approach for spatial transmission and heterogeneity in the COVID-19 propagation

L. Roques[1], O. Bonnefon[1], V. Baudrot[1], S. Soubeyrand[1] and H. Berestycki[2,3]

[1]INRAE, BioSP, 84914 Avignon, France
[2]EHESS, CNRS, CAMS, Paris, France
[3]Senior Visiting fellow, HKUST Jockey Club Institute for Advanced Study, Hong Kong University of Science and Technology, Hong Kong

LR, 0000-0001-6647-7221; VB, 0000-0003-1327-9728;
SS, 0000-0003-2447-3067; HB, 0000-0003-1724-2279

Raw data on the number of deaths at a country level generally indicate a spatially variable distribution of COVID-19 incidence. An important issue is whether this pattern is a consequence of environmental heterogeneities, such as the climatic conditions, during the course of the outbreak. Another fundamental issue is to understand the spatial spreading of COVID-19. To address these questions, we consider four candidate epidemiological models with varying complexity in terms of initial conditions, contact rates and non-local transmissions, and we fit them to French mortality data with a mixed probabilistic-ODE approach. Using statistical criteria, we select the model with non-local transmission corresponding to a diffusion on the graph of counties that depends on the geographic proximity, with time-dependent contact rate and spatially constant parameters. This suggests that in a geographically middle size centralized country such as France, once the epidemic is established, the effect of global processes such as restriction policies and sanitary measures overwhelms the effect of local factors. Additionally, this approach reveals the latent epidemiological dynamics including the local level of immunity, and allows us to evaluate the role of non-local interactions on the future spread of the disease.

## 1. Introduction

In France, the first cases of COVID-19 epidemic were reported on 24 January 2020 [1], although it appeared later that some cases were already present in December 2019 [2]. Since then, the first important clusters were observed in February in the Grand Est

region and the Paris region. A few months later, at the beginning of June, the spatial pattern of the disease spread seems to have kept track with these first introductions [3]. As this spatial pattern may also be correlated with covariates such as climate [4] (see also electronic supplementary material, S1), a fundamental question is to assess whether this pattern is the consequence of a heterogeneous distribution of some covariates or if it can be explained by the heterogeneity of the initial introduction points. In the latter case, we want to know if the epidemic dynamics simply reflects this initial heterogeneity and can be modelled without taking into account any spatial heterogeneity in the local conditions.

SIR epidemiological models and their extensions have been proposed to study the spread of the COVID-19 epidemic at the country or state scale (e.g. [5] in France and [6] in three US states), and at the regional scale ([7,8] in China, [3] in France and [9] in Italy). In all cases, a different set of parameters has been estimated for each considered region/province. One of the main goals of our study is to check whether the local mortality data at a thin spatial scale can still be well explained by a single set of parameters, at the country scale, but spatially varying initial conditions. In this study, we consider SIR models with time-dependent contact rate, to track changes over time in the dynamic reproductive number as in the branching process considered in [6]. Using standard statistical criteria, we compare four types of models, whose parameters are either global at the country scale or spatially heterogeneous, and with non-local transmission or not. We work with mortality data only, as these data appear to be more reliable and less dependent on local testing strategies than confirmed cases, in settings where cause of death is accurately determined [10]. Additionally, it was already shown that for this type of data, SIR models outperform other classes of models (an SEIR models and a branching process) in the three US states considered in [6]. The approach we develop here is in line with the general principle of using parsimonious models eloquently emphasized in [6].

It is widely accepted that the age structure of the population [11] and the presence of asymptomatic infectious individuals [12] affect the dynamics of the COVID-19 epidemic. Several works take these features into account (e.g. [13,14] for models with several age classes and [15] for models with a specific compartment of unreported cases). However, these approaches bring in a new difficulty in that they involve a larger number of unknown parameters leading to identifiability issues and underdetermination of the models by the data. As is often the case, there is a trade-off between searching for a more realistic description and models that can be trusted because they still match the data satisfactorily with a small set of parameters to identify. From this point of view, we observe that more parsimonious approaches, in which the above-mentioned compartments (e.g. reported and unreported cases) are merged into a single 'infectious' compartment, are still able to capture key features of the epidemic directly from surveillance data, in some cases even earlier than more complex approaches. For instance, with a simple SIRD model coupled with a probabilistic observation model, [5] estimated the infection fatality ratio (IFR) at an early stage of the epidemic, and obtained results that matched subsequent analyses based on more detailed and realistic models such as in the study of Institut Pasteur [3] that involved several age classes and a very precise description of the transition from hospital admission to ICU and death. It turned out that in both cases, the authors obtained exactly the same IFR of 0.5% (excluding deaths in nursing homes). Yet, the simple SIRD model hinged on a reduced dataset (number of tests, number of positive cases and number of deaths in France) as compared to the study [3], which required additional data in France and also data from the *Diamond Princess* cruise ship. Thus, Occam's razor leads us here to give preference to the simplest model whenever it is sufficient to fit the data.

Another important issue that such models may help to address is the quantification of the relative effects of restrictions on inter-regional travel, versus reductions in the probability of infection per contact at the local scale. France went into lockdown on 17 March 2020, which was found to be very effective in reducing the spread of the disease. It divided the effective reproduction number (number $\mathcal{R}_t$ of secondary cases generated by an infectious individual [16]) by a factor 5–7 at the country scale by 11 May [3,17]. This is to be compared with the estimate of the basic reproduction number $\mathcal{R}_0$ carried out in France at the early onset of the epidemic, before the country went into lockdown (with values $\mathcal{R}_0 = 2.96$ in [3] and $\mathcal{R}_0 = 3.2$ in [5]). Contact-surveys data [18] for Wuhan and Shanghai, China, have found comparable estimates of the reduction factor. The national lockdown in France induced important restrictions on movement, with e.g. a mandatory home confinement except for essential journeys, leading to a reduction of the number of contacts. In parallel, generalized mask wearing and use of hydroalcoholic gels reduced the probability of infection per contact.

After the lockdown, restrictions policies were generally based on raw data rather than modelling. An efficient regulation at the local scale would need to know the current number of infectious and the level of immunity at the scale of counties. The French territory is divided into administrative units called 'départements', analogous to counties. We use 'counties' hereafter for départements. These quantities

cannot be observed directly in the absence of large-scale testing campaigns or spatial random sampling. In particular, there is a large number of unreported cases. Previous studies developed a mixed mechanistic–probabilistic framework to estimate these quantities at the country scale. These involved estimating the relative probability of getting tested for an infected individual versus a healthy individual, leading to a factor ×8 between the number of confirmed cases and the actual number of cases before the lockdown [5], and ×20 at the end of the lockdown period [17]. This type of framework—often referred to as 'mechanistic-statistical modelling'—aims at connecting the solution of continuous state models such as differential equations with complex data, such as noisy discrete data, and identifying latent processes such as the epidemiological process under consideration here. Initially, introduced for physical models and data [19], it is becoming standard in ecology [20].

Using this framework, the objectives of our study are (i) to assess whether the spatial pattern observed in France is due to some local covariates or is simply the consequence of the heterogeneity in the initial conditions together with global processes at the country scale, (ii) to evaluate the role of non-local interactions on the spread of the disease, and (iii) to propose a tool for real-time monitoring the main components of the disease in France, with a particular focus on the local level of immunity.

# 2. Material and methods

## 2.1. Data

Mainland France (excluding Corsica island) is made of 94 counties called 'départements'. The daily number of hospital deaths—excluding nursing homes—at the county scale are available from Santé Publique France since 18 March 2020 (and available as electronic supplementary material). The daily number of observed deaths (still excluding nursing homes) in county $k$ during day $t$ is denoted by $\hat{\mu}_{k,t}$.

We denote by $[t_i, t_f]$ the observation period and by $n_d$ the number of considered counties. To avoid a too large number of counties with 0 deaths at initial time, the observation period ranges from $t_i = 30$ March to $t_f = 11$ June, corresponding to $n_t = 74$ days of observation. All the counties of mainland France (excluding Corsica) are taken into account, but the Ile-de-France region, which is made of eight counties with a small area is considered as a single geographic entity. This leads to $n_d = 87$.

## 2.2. Mechanistic-statistical framework

The mechanistic-statistical framework is a combination of a mechanistic model that describes the epidemiological process, a probabilistic observation model and a statistical inference procedure. We begin with the description of four mechanistic models, whose main characteristics are given in table 1.

### 2.2.1. Mechanistic models

*Model $\mathcal{M}_0$: SIR model for the whole country.* The first model is the standard mean field SIRD model that was used in [5,17]

$$\left.\begin{array}{l} S'(t) = -\dfrac{\alpha(t)}{N}\,S\,I, \\[2mm] I'(t) = \dfrac{\alpha(t)}{N}\,S\,I - (\beta + \gamma)\,I, \\[2mm] R'(t) = \beta I \\[2mm] D'(t) = \gamma I, \end{array}\right\} \tag{2.1}$$

and

with $S$ the susceptible population, $I$ the infectious population, $R$ the recovered population, $D$ the number of deaths due to the epidemic and $N$ the total population, in the whole country. For simplicity, we assume that $N$ is constant, equal to the current population in France, thereby neglecting the effect of the small variations of the population on the coefficient $\alpha(t)/N$. The parameter $\alpha(t)$ is the contact rate (to be estimated) and $1/\beta$ is the mean time until an infectious becomes recovered. The results in [21] show that the median period of viral shedding is 20 days, but the infectiousness tends to decay before the end of this period: the results in [22] indicate that infectiousness starts from 2.5 days before symptom onset and declines within 7 days of illness onset. Based on these observations, we assume here that $1/\beta = 10$ days. The parameter $\gamma$ corresponds to the (hospital) death rate of the infectious. The infection fatality ratio (IFR) was estimated in France by several studies [3,5,17], leading in each case to a value

**Table 1.** Main characteristics of the four models. The quantity $n_t = 74$ corresponds to the number of days of the observation period and $n_d = 87$ corresponds to the number of administrative units.

|  | heterog. initial data | heterog. contact rate | intercounty transmission | no. parameters |
|---|---|---|---|---|
| $\mathcal{M}_0$ | no | no | no | $n_t$ |
| $\mathcal{M}_1$ | yes | no | no | $n_t$ |
| $\mathcal{M}_2$ | yes | yes | no | $n_d \times n_t$ |
| $\mathcal{M}_3$ | yes | no | yes | $n_t + 2$ |

0.5%. This value is consistent with previous findings in China (0.66%) [23] and Germany (0.36%) [24], is lower than in Spain (1.15%) [25] and on the *Diamond Princess* cruise ship (1.3%) [26]. The value 0.5% in France is probably underestimated as it only takes into account the deaths at hospital, but is consistent with the data that we used here (hospital mortality). The IFR corresponds to the fraction of the infected who die, that is: $\gamma/(\gamma + \beta)$. Thus, a value of 0.5% for the IFR implies a value $\gamma = 5 \times 10^{-4}$.

*Model $\mathcal{M}_1$: SIR model at the county scale with globally constant contact rate and no spatial transmission.* The model $\mathcal{M}_0$ is applied at the scale of each county $k$, leading to compartments $S_k, I_k, R_k, D_k$ that satisfy an equation of the form (2.1), with $N$ replaced by $N_k$, the total population in the county $k$. In this approach, the contact rate $\alpha(t)$ is assumed to be the same in all of the counties.

*Model $\mathcal{M}_2$: SIR model at the county scale with spatially heterogeneous contact rate and no spatial transmission.* With this approach, the model $\mathcal{M}_1$ is extended by assuming that the contact rate $\alpha_k(t)$ depends on the considered county.

*Model $\mathcal{M}_3$: County scale model with globally constant contact rate and spatial transmission.* The model $\mathcal{M}_1$ is extended to take into account disease transmission events between the counties

$$
\left.
\begin{aligned}
S'_k(t) &= -\frac{\rho(t)}{N_k} S_k \sum_{j=1}^{n_d} w_{j,k} I_j, \\
I'_k(t) &= \frac{\rho(t)}{N_k} S_k \sum_{j=1}^{n_d} w_{j,k} I_j - (\beta + \gamma) I_k, \\
R'_k(t) &= \beta I_k \\
D'_k(t) &= \gamma I_k.
\end{aligned}
\right\}
\tag{2.2}
$$

and

The parameter $\rho(t)$ is a 'local' contact rate, i.e. the contact rate corresponding to the interactions between the susceptible individuals of the county $k$ with the infectious individuals from the same county. The weights $w_{j,k}$ are dimensionless and describe the ratios between the contact rates associated with the $j$–$k$ interactions and those associated with the $k$–$k$ interactions. We assume a power law decay of these weights with the distance

$$
w_{j,k} = \frac{1}{1 + (\mathrm{dist}(j, k)/d_0)^\delta},
\tag{2.3}
$$

with $\mathrm{dist}(j, k)$ the geographical distance (in km) between the centroids of counties $j$ and $k$, $d_0 > 0$ a proximity scale, and $\delta > 0$. Thus, the model involves two new global parameters, $d_0$ and $\delta$, compared to model $\mathcal{M}_1$. This model extends the Kermack–McKendrick SIR model to take into account non-local spatial interactions. It was introduced by Kendall [27] in continuous variables. The model we adopt here is inspired by the study of Bonnasse-Gahot *et al.* [28] in a different context where the same types of weights have been used. We thus take into account diffusion on the weighted graph of counties in France. This amounts to considering that individuals in a given county are infected by individuals from other counties with a probability that decreases with distance as a power law, in addition to contagious individuals from their own county. This dependence of social spatial interactions with respect to the distance is supported, notably, by Brockmann *et al.* [29], who analysed the short-time dispersal of bank notes in the USA. This type of power-law decay is commonly used in 'gravity models' to describe the flow of people from one area to another [30]. Though promising extensions have been proposed, with for instance higher-order interactions between cities [31], this type of interactions remains a standard choice in infectious disease modelling (e.g. [32–34]). We also refer to [35] for a thorough discussion on the various applications of power-law dispersal kernels since they were introduced by Pareto [36].

With this non-local contagion model, in contradistinction to epidemiological models with dispersion such as reaction–diffusion epidemiological models [37], the movements of the individuals are not modelled explicitly. The model implicitly assumes that infectious individuals may transmit the disease to susceptible individuals in other counties, but eventually return to their county of origin. This has the advantage of avoiding unrealistic changes in the global population density.

### 2.2.2. Observation model

We denote by $\bar{D}_k(t)$ the expected cumulative number of deaths given by the model, in county $k$. With the mean-field model $\mathcal{M}_0$, we assume that it is proportional to the population size: $\bar{D}_k(t) = D(t)\,N_k/N$, with $N_k$ the population in county $k$ and $N$ the total French population. With models $\mathcal{M}_1$, $\mathcal{M}_2$ and $\mathcal{M}_3$, we simply have $\bar{D}_k(t) = D_k(t)$. The expected daily increment in the number of deaths given by the models in a county $k$ is $\bar{D}_k(t) - \bar{D}_k(t-1)$.

The observation model assumes that the daily number of new observed deaths $\hat{\mu}_{k,t}$ in county $k$ follows a Poisson distribution with mean value $\bar{D}_k(t) - \bar{D}_k(t-1)$

$$\hat{\mu}_{k,t} \sim \mathrm{Poisson}(\bar{D}_k(t) - \bar{D}_k(t-1)). \tag{2.4}$$

Note that the time $t$ in the mechanistic models is a continuous variable, while the observations $\hat{\mu}_{k,t}$ are reported at discrete times. For the sake of simplicity, we used the same notation $t$ for the days in both the discrete and continuous cases. In formula (2.4), $\bar{D}_k(t)$ (resp. $\bar{D}_k(t-1)$) is computed at the end of day $t$ (resp. $t-1$).

## 2.3. Initial conditions

In models $\mathcal{M}_1$, $\mathcal{M}_2$ and $\mathcal{M}_3$, at initial time $t_i$, we assume that the number of susceptible cases is equal to the number of inhabitants in county $k$: $S_k(t_i) = N_k$, the number of recovered is $R(t_i) = 0$ and the number of deaths is given by the data: $D_k(t_i) = \hat{\mu}_{k,t_i}$. To initialize the number of infectious, we use the equation $D'(t) = \gamma\,I(t)$, and we define $I(t_i)$ as $1/\gamma \times$ (mean number of deaths over the period ranging from $t_i$ to 29 days after $t_i$)

$$I_k(t_i) = \frac{1}{\gamma}\frac{1}{30}\sum_{s=t_i,\dots,t_i+29}\hat{\mu}_{k,s}. \tag{2.5}$$

The 30-days window was chosen such that there was at least one infectious case in each county. In model $\mathcal{M}_0$, the initial conditions are obtained by adding the initial conditions of model $\mathcal{M}_1$ (or equivalently, $\mathcal{M}_2$) over all the counties.

## 2.4. Statistical inference

### 2.4.1. Real-time monitoring of the parameters and data assimilation procedure

To smooth out the effect of small variations in the data, and to avoid identifiability issues due to the large number of parameters, while keeping the temporal dependence of the parameters, the parameters $\alpha(t)$ and $\alpha_k(t)$ of the ODE models $\mathcal{M}_0$, $\mathcal{M}_1$ and $\mathcal{M}_2$ are estimated by fitting auxiliary problems with time-constant parameters over moving windows $(t-\tau/2,\, t+\tau/2)$ of fixed duration equal to $\tau$ days. These auxiliary problems are denoted, respectively, by $\tilde{\mathcal{M}}_{0,t}$, $\tilde{\mathcal{M}}_{1,t}$ and $\tilde{\mathcal{M}}_{2,t}$ (see electronic supplementary material S2 for a precise formulation of these problems). The initial conditions associated with this system, at the date $t-\tau/2$ are computed iteratively from the solution of $\mathcal{M}_0$, $\mathcal{M}_1$ and $\mathcal{M}_2$, respectively.

### 2.4.2. Inference procedure

For simplicity, in all cases, we denote by

$$f_{\bar{D}_k,\hat{\mu}_k}(s) := \frac{(\bar{D}_k(s) - \bar{D}_k(s-1))^{\hat{\mu}_{k,s}}}{\hat{\mu}_{k,s}!}\,\mathrm{e}^{-(\bar{D}_k(s)-\bar{D}_k(s-1))}$$

the probability mass function associated with the observation process (2.4) at date $s$ in county $k$, given the expected cumulative number of deaths $\bar{D}_k$ given by the considered model in county $k$.

In models $\tilde{\mathcal{M}}_{0,t}$ and $\tilde{\mathcal{M}}_{1,t}$, the estimated parameter is $\tilde{\alpha}$. The likelihood $\mathcal{L}$ is defined as the probability of the observations (here, the increments $\{\hat{\mu}_{k,s}\}$) conditionally on the parameter. Using the assumption

that the increments $\hat{\mu}_{k,s}$ are independent conditionally on the underlying SIRD process $\tilde{\mathcal{M}}_{0,t}$ (resp. $\tilde{\mathcal{M}}_{1,t}$), we get

$$\mathcal{L}(\tilde{\alpha}) := P\left(\left\{\hat{\mu}_{k,s}, \; k = 1, \ldots, n_d, \; s = t - \frac{\tau}{2}, \ldots, t + \frac{\tau}{2}\right\} | \tilde{\alpha}\right)$$

$$= \prod_{k=1}^{n_d} \prod_{s=t-\tau/2}^{t+\tau/2} f_{\bar{D}_k, \hat{\mu}_k}(s).$$

We denote by $\tilde{\alpha}_t^*$ the corresponding maximum-likelihood estimator, and we set $\alpha(t) = \tilde{\alpha}_t^*$ in model $\mathcal{M}_0$ (resp. $\mathcal{M}_1$).

For model $\tilde{\mathcal{M}}_{2,t}$, the inference of the parameters $\tilde{\alpha}_k$ is carried out independently in each county, leading to the likelihoods

$$\mathcal{L}_k(\tilde{\alpha}_k) := P\left(\left\{\hat{\mu}_{k,s}, \; s = t - \frac{\tau}{2}, \ldots, t + \frac{\tau}{2}\right\} | \tilde{\alpha}_k\right)$$

$$= \prod_{s=t-\tau/2}^{t+\tau/2} f_{\bar{D}_k, \hat{\mu}_k}(s).$$

We denote by $\tilde{\alpha}_{k,t}^*$ the corresponding maximum-likelihood estimator, and we set $\alpha_k(t) = \tilde{\alpha}_{k,t}^*$ in model $\mathcal{M}_2$.

For model $\mathcal{M}_3$, we apply a two-stage estimation approach. We first use the estimate obtained with model $\mathcal{M}_1$ by setting $\rho(t) = C\,\alpha(t)$, where $\alpha(t)$ is the estimated contact rate of model $\mathcal{M}_1$ and $C \in (0, 1)$ is a constant (to be estimated; note that estimating $\rho(t)$ means estimating $n_t$ parameters). This constant $C$ can be interpreted as the fraction of the overall contact rate $\alpha(t)$ which can be attributed to within-county transmissions. Given $\alpha(t)$, the only parameters to be estimated are the constant $C$, the proximity scale $d_0$ and the exponent $\delta$. They are estimated by maximizing

$$\mathcal{L}(C, d_0, \delta) := P(\{\hat{\mu}_{k,s}, \; k = 1, \ldots, n_d, \; s = t_i, t_f\} | C, d_0, \delta)$$

$$= \prod_{k=1}^{n_d} \prod_{s=t_i}^{t_f} f_{\bar{D}_k, \hat{\mu}_k}(s).$$

### 2.4.3. Model selection

We use the Akaike information criterion (AIC, [38]) and the Bayesian information criterion (BIC, [39]) to compare the models. For both criteria, we need to compute the likelihood function associated with the model, with the parameters determined with the above inference procedure

$$\mathcal{L}(\mathcal{M}_m) = \prod_{k=1}^{n_d} \prod_{s=t_i}^{t_f} f_{\bar{D}_k, \hat{\mu}_k}(s), \tag{2.6}$$

with $m = 0, 1, 2, 3$ and $\bar{D}_k$ the expected (cumulative) number of deaths in county $k$ given by model ($\mathcal{M}_m$). Given the number of parameters $\sharp \mathcal{M}_m$ estimated in model $\mathcal{M}_m$, the AIC score is defined as follows:

$$AIC(\mathcal{M}_m) = 2\,\sharp \mathcal{M}_m - 2\,\ln\left(\mathcal{L}(\mathcal{M}_m)\right),$$

and the BIC score:

$$BIC(\mathcal{M}_m) = \sharp \mathcal{M}_m \ln\left(K\right) - 2\,\ln\left(\mathcal{L}(\mathcal{M}_m)\right),$$

with $K = n_d \times n_t = 6438$ the number of data points.

### 2.4.4. Numerical methods

To find the maximum-likelihood estimator, we used a Broyden–Fletcher–Goldfarb–Shanno (BFGS) constrained minimization algorithm, applied to $-\ln(\mathcal{L})$ via the Matlab® function *fmincon*. The ODEs were solved thanks to a standard numerical algorithm, using Matlab® *ode45* solver.

# 3. Results

## 3.1. Model fit and model selection

To assess model fit, we compared the daily increments in the number of deaths given by each of the four models with the data. For each model, we used the parameters corresponding to the maximum-likelihood estimators. The comparisons are carried out at the regional scale: mainland France is

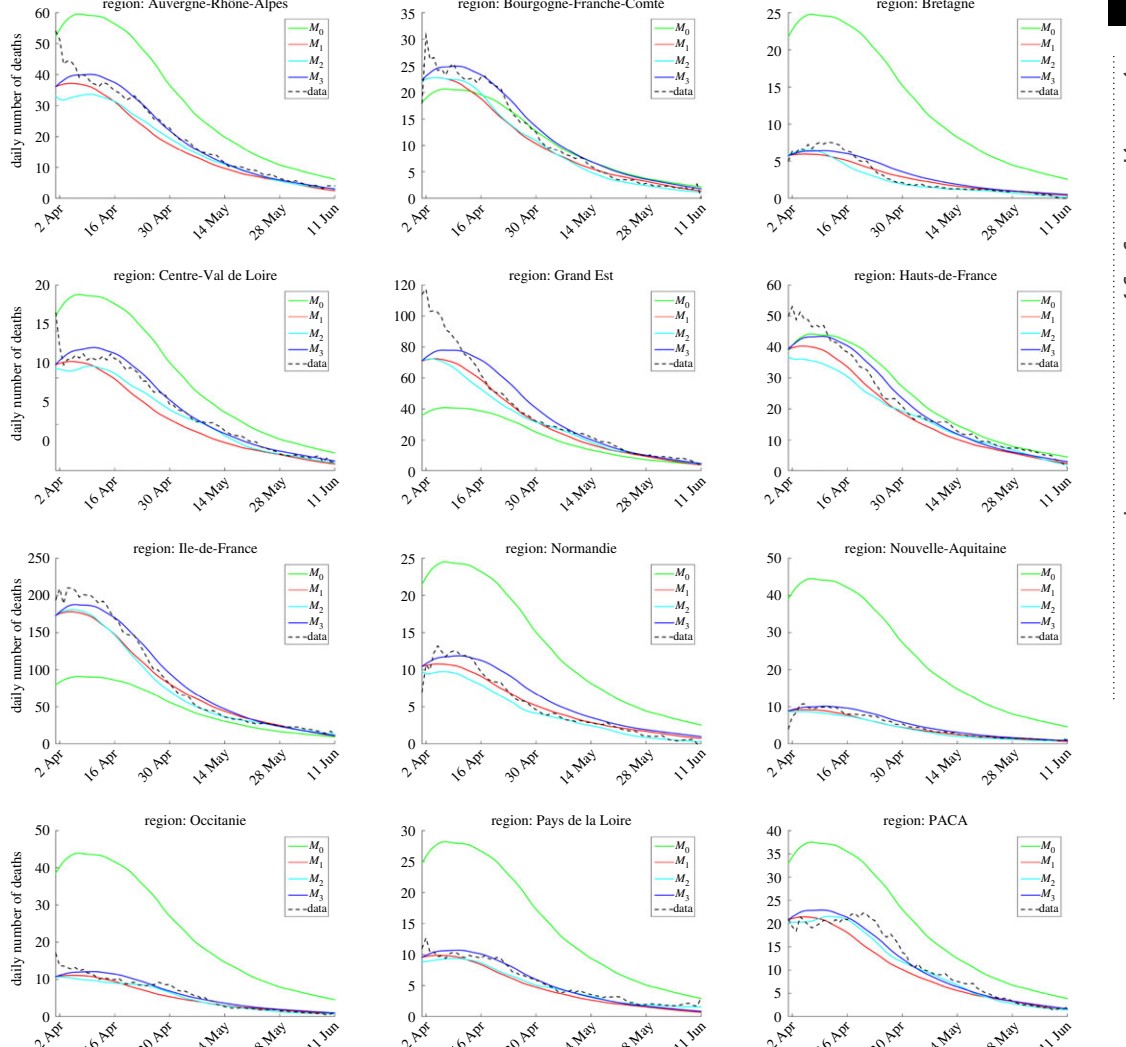

**Figure 1.** Model fit at the regional scale. The data have been smoothed (moving average over 15 days), for easier graphical comparison with the models.

divided into 12 administrative regions, each of which is made of several counties. The results are presented in figure 1. In all the regions, the models $\mathcal{M}_1$, $\mathcal{M}_2$ and $\mathcal{M}_3$ lead to a satisfactory visual fit of the data, whereas the mean field model $\mathcal{M}_0$ does not manage to reproduce the variability of the dynamics among the regions.

The log-likelihood, AIC and BIC values are given in table 2. Models $\mathcal{M}_1$, $\mathcal{M}_2$ and $\mathcal{M}_3$ lead to significantly higher likelihood values than model $\mathcal{M}_0$. This reflects the better fit obtained with these three models, compared to model $\mathcal{M}_0$ and shows the importance of taking into account the spatial heterogeneities in the initial densities of infectious cases. On the other hand, the log-likelihood, though higher with model $\mathcal{M}_2$ is close to that obtained with $\mathcal{M}_1$, and the model selection criteria are both strongly in favour of model $\mathcal{M}_1$. This shows that the spatial heterogeneity in the contact rate does not have a significant effect on the epidemic dynamics within mainland France.

Model $\mathcal{M}_3$ with spatial transmission leads to an intermediate likelihood value, between those of models $\mathcal{M}_1$ and $\mathcal{M}_2$, with only two additional parameters with respect to model $\mathcal{M}_1$. As a consequence, the model selection criteria exhibit strong evidence in favour of the selection of model $\mathcal{M}_3$. This means a large part of the difference between models $\mathcal{M}_1$ and $\mathcal{M}_2$ can be captured by taking into account the spatial transmission, which therefore seems to have a significant effect on the epidemic dynamics. An important advantage in using statistical criteria such as AIC and BIC is that they were conceived to avoid over-parametrization, and to select as it were the 'true process' that generated the data [40]. From a pragmatic viewpoint, the notion of 'true process' has to be understood as 'likely process'. Since these criteria only provide us with a model comparison, they only yield specific conclusions. Here, they allow

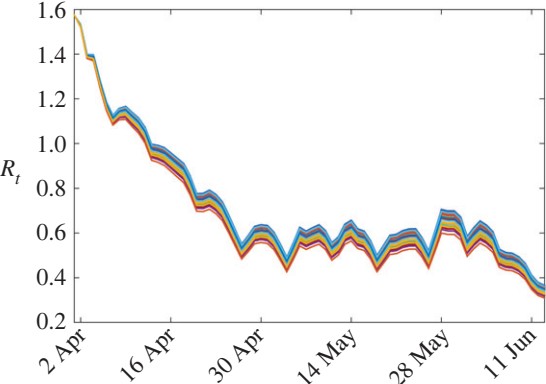

**Figure 2.** Dynamics of the effective reproduction rate $\mathcal{R}_t^k$ given by model $\mathcal{M}_1$ over the 87 considered counties.

**Table 2.** Log-likelihood, AIC and BIC values for the four models. The last column ΔAIC corresponds to the difference with the AIC value of the best model (here $\mathcal{M}_3$).

| model | AIC | BIC | log-likelihood | ΔAIC |
|---|---|---|---|---|
| $\mathcal{M}_0$ | $2.68 \times 10^4$ | $2.73 \times 10^4$ | $-13.4 \times 10^3$ | $-9.57 \times 10^3$ |
| $\mathcal{M}_1$ | $1.74 \times 10^4$ | $1.79 \times 10^4$ | $-8.62 \times 10^3$ | $-220$ |
| $\mathcal{M}_2$ | $2.97 \times 10^4$ | $7.36 \times 10^4$ | $-8.41 \times 10^3$ | $-1.25 \times 10^4$ |
| $\mathcal{M}_3$ | $1.72 \times 10^4$ | $1.77 \times 10^4$ | $-8.52 \times 10^3$ | $0$ |

us to state that global processes are more likely to have shaped the spatial distribution than local covariates ($\mathcal{M}_1$ versus $\mathcal{M}_2$). Indeed, introducing a local effect does not improve the AIC and BIC values. Furthermore, intercounty transmission has played a significant role in the spread of the epidemics ($\mathcal{M}_1$ versus $\mathcal{M}_3$).

As a by-product of the estimation of the parameter $\alpha(t)$ (resp. $\alpha_k(t)$) of model $\mathcal{M}_1$ (resp. $\mathcal{M}_2$), we get an estimate of the effective reproduction number in each county, which is given by the formula [41]

$$\mathcal{R}_t^k = \frac{\alpha(t)}{\beta + \gamma} \frac{S_k(t)}{N_k},$$

so that $I_k'(t) < 0$ (the epidemic tends to vanish) whenever $\mathcal{R}_t^k < 1$, whereas $I_k'(t) > 0$ whenever $\mathcal{R}_t^k > 1$ (the number of infectious cases in the population follows an increasing trend). The dynamics of $\mathcal{R}_t^k$ obtained with model $\mathcal{M}_1$ are depicted in figure 2, clearly showing a decline in $\mathcal{R}_t^k$, as already observed in [3,17], but there the computation was at a fixed date.

For model $\mathcal{M}_3$, the maximum-likelihood estimation gives $C = 0.87$, $d_0 = 2.16$ km and $\delta = 1.85$, which yields a nearly quadratic decay of the weights with the distance. The value of $d_0$, indicates that non-local contagion plays a secondary role compared to within-county contagion: the minimum distance between two counties is 36 km, leading to a weight of 5.5/1000, to be compared with the weight 1 for within-county contagion. However, the fact that the parameter $C$ is significantly smaller than 1 (recall that the contact rate in model $\mathcal{M}_3$ is $\rho(t) = C \alpha(t)$ with $\alpha(t)$ the contact rate in model $\mathcal{M}_2$) shows that the non-local contagion term plays an important role on the spreading of the epidemic.

### 3.1.1. Immunity rate

Using model $\mathcal{M}_3$, which leads to the best fit, we derive the number of recovered individuals (considered here as immune) at a date $t$, in each county, and the immunity rate $R_k/N_k$. It is presented at time $t_f$ in figure 3. The full timeline of the dynamics of immunity obtained with model $\mathcal{M}_3$ since the beginning of April is available as electronic supplementary material (see SI 1).

### 3.1.2. Limiting movement versus limiting the probability of transmission per contact

Before the lockdown, the basic reproduction number $\mathcal{R}_0$ in France was about 3 [3,5], and was then reduced by a factor 5–7, leading to values around 0.5 (see [3,17] and figure 2). This corresponds to a

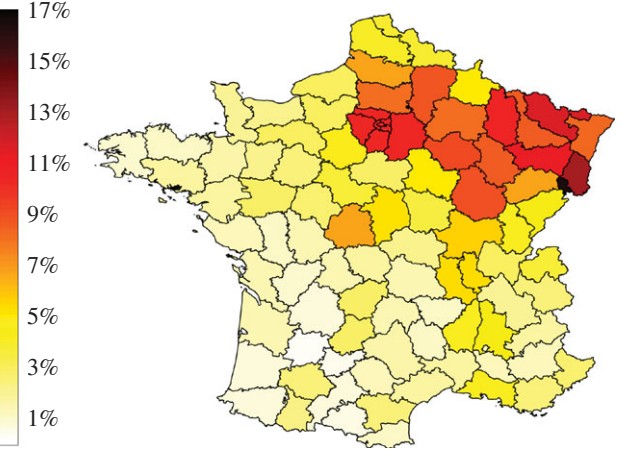

**Figure 3.** Estimated immunity rate, at the county scale, by 11 June 2020, using model $\mathcal{M}_3$.

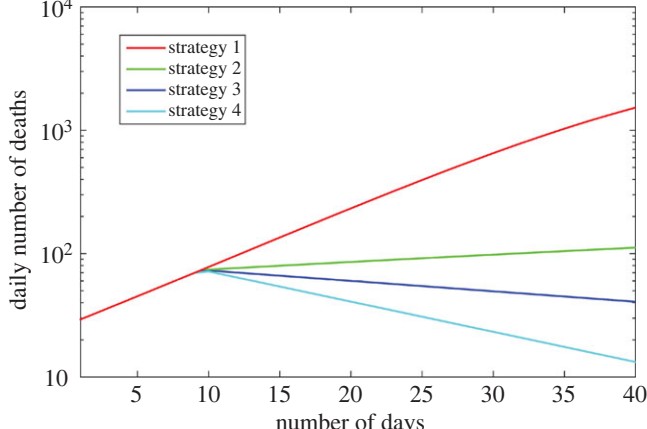

**Figure 4.** Daily number of deaths due to a new outbreak in logarithmic scale; comparison between four management strategies. The number of deaths is computed over the whole country.

contact rate $\alpha(t) \approx \beta \mathcal{R}_t \approx 0.3$ before the lockdown and $\alpha(t) \approx 0.05$ after the lockdown in model $\mathcal{M}_1$. Let us now consider a hypothetical scenario of a new outbreak with an effective reproduction number that rises again to reach values above 1. Due to the higher awareness of the population with respect to epidemic diseases and the new sanitary behaviours induced by the first COVID-19 wave, the reproduction number will probably not reach again values as high as 3.

The new outbreak scenario is described as follows: we start from the state of the epidemic at 11 June, and we assume a 'local' contact rate $\rho(t)$ jumping to the value 0.11 in model $\mathcal{M}_3$ (corresponding to a twofold increase compared to the previous 30 days). In parallel, to describe the lifting of restrictions on individual movements, we set $d_0 = 20$ km for the proximity scale in model $\mathcal{M}_3$. This new outbreak runs during 10 days, and then, we test four strategies:

— Strategy 1: no restriction. The parameters remain unchanged: $\rho(t) = 0.11$ and $d_0 = 20$ km.
— Strategy 2: restriction on intercounty movement. The parameter $\rho(t) = 0.11$ is unchanged, but $d_0 = 2.16$ km, corresponding to its estimated value during the period $(t_i, t_f)$.
— Strategy 3: reduction of the contact rate within each county (e.g. by wearing masks), but no restriction on intercounty movement: $\rho(t) = 0.05$ and $d_0 = 20$ km.
— Strategy 4: reduction of the contact rate within each county and restriction on intercounty movement: $\rho(t) = 0.05$ and $d_0 = 2.16$ km.

The daily number of deaths corresponding to each scenario is presented in figure 4. We estimate in each case a value of the effective reproduction number $\mathcal{R}_t$ over the whole country by fitting the global number of infectious cases with an exponential function over the last 30 days. As expected, the more restrictive

the strategy, the less the number of deaths. After 30 days, the cumulative number of deaths with the first strategy is 17 271, and $\mathcal{R}_t \approx 2$. Restriction on intercounty movement (strategy 2) leads to a 81% decrease in the cumulative number of deaths (3281 deaths) and $\mathcal{R}_t \approx 1.2$; reducing the contact rate within each county leads to a 88% decrease (strategy 3, 2139 deaths) and $\mathcal{R}_t \approx 0.8$; finally, control strategy 4, which combines both types of restrictions leads to a 91% decrease (1503 deaths) and $\mathcal{R}_t \approx 0.4$.

# 4. Discussion

We show here that a parsimonious model can reproduce the local dynamics of the COVID-19 epidemic in France with a relatively high goodness of fit. This is achieved despite the spatial heterogeneity across French counties of some environmental factors potentially influencing the disease propagation. Indeed, our model only involves the initial spatial distribution of the infectious cases and spatially homogeneous (i.e. countrywide) parameters. For instance, we do observe a negative correlation between the mean temperature and the immunity rate (see electronic supplementary material, figure S1); however, it does not reflect a causality. Actually, our study shows that if there is an effect of local covariates such as the mean temperature or e.g. the local age structure, on the spread of the disease once it emerged, its effect is of lower importance compared to the global processes at work at the country scale, such as sanitary measures. Local covariates might play a major role in the emergence of the disease, but our work focuses on the disease dynamics after the emergence.

Hence, we find that initial conditions and spatial diffusion are likely to be the main drivers of the spatial pattern of the COVID-19 epidemic. This result may rely on specific circumstances: e.g. mainland France covers a relatively middle-sized area, with mixed urban and countryside populations across the territory, a relatively homogeneous population age distribution, and a high level of centralism for public decision (in particular regarding the disease-control strategy). Of course, these features are not universal. In other countries with more socio-environmental diversity within which environmental drivers and state decentralization could significantly induce spatial variations in disease spread and, consequently, in which countrywide parameters would not be appropriate. Moreover, at the global scale, the COVID-19 dynamics in different countries seem highly contrasted as illustrated by the data of Johns Hopkins University Center for Systems Science and Engineering [42]— see also http://covid19-forecast.biosp.org/ [43]—and could probably not be explained with a unique time-varying contact rate parameter. Nonetheless, this model could be adapted to many other situations at an appropriate geographical level, with a single well-defined political decision process.

Back-of-the-envelope estimates suggest that herd immunity requires that a fraction of approximately $1 - 1/\mathcal{R}_0 \approx 70\%$ of the population has been infected. It is far from being reached at the country scale in France, but we observe that the fraction of immune individuals strongly varies across the territory, with possible local immunity effects. For instance, in the most impacted county, our mathematical model's best estimate is that the immunity rate is 16%, whereas it is less than 1% in less affected counties. At a thinner grain scale, even higher rates may be observed, for instance, by 4 April the proportion of people with confirmed SARS-CoV-2 infection based on antibody detection was 41% in a high-school located in Northern France [44].

Real-time monitoring of the immunity level will be crucial to define efficient management policies, if a new outbreak occurs. We propose such a tool which is based on the modelling approach $\mathcal{M}_3$ of this paper (see Immunity tab in http://covid19-forecast.biosp.org/). Remarkably, the estimated levels of immunity are comparable to those observed in Spain by population-based serosurveys [45], with values ranging from 0.5% to 13.0% at the beginning of May at the provincial resolution. Actually, a very recent study [46] (October 2020) provides estimates for prevalence of anti-SARS-CoV-2 antibodies in France. It shows that the nationwide seroprevalence was 4.93% ([4.02–5.89]) by mid-May. This is consistent with the immunity rate given by our model at the same dates (4.51%). The distribution of the immunity rate in figure 3 here also seems in good qualitative agreement with the findings of this study.

Our results indicate that, at least in the simple mathematical model that we considered here, travel restrictions alone, although they may have a significant effect on the cumulative number of deaths and the reproduction number over a definite period, are less efficient—at the country scale—in reducing transmission than social distancing and other sanitary measures. Obviously, these results may strongly depend on the parameter values, which have been chosen here on the basis of values estimated during the lockdown period. This is consistent with results for China [47], where travel restrictions to and from Wuhan have been shown to have a modest effect unless paired with other public health interventions and behavioural changes.

The model selection criteria led to strong evidence in favour of the selection of model $\mathcal{M}_3$ with non-local transmission and spatially constant contact rate. It is much more parsimonious than the fully heterogeneous model $\mathcal{M}_2$ and is therefore better suited to isolating key features of the epidemiological dynamics [6]. Despite important restrictions on movement during the considered period (mandatory home confinement except for essential journeys until 11 May and a 100 km travel restriction until 2 June), the model $\mathcal{M}_3$ was also selected against the model $\mathcal{M}_1$ which does not take into account non-local transmission. This shows that intercounty transmission is one of these key features that the non-local model manages to take into account.

More generally, in France just as in Italy [9], the spatial pattern of COVID-19 incidence indicates that spatial processes play a key role. At this stage, only a few models can address this aspect. Some have adopted a detailed spatio-temporal modelling approach and use mobility data (see [9] for an SEIR-like model with nine compartments). The framework we develop here, including the non-local model and the associated estimation procedure, should be of broad interest in studying the spatial dynamics of epidemics, due to its theoretical and numerical simplicity and its ability to accurately track the epidemics. This approach applies when geographical distance matters, which may not be the case at the scale of countries like the USA. However, it does at more regional scales. Furthermore, we can envision natural extensions of the approach that would take into account long-range dispersal events in the interaction term.

We considered here a macroscopic deterministic framework. That is, we used continuous models at the population scale rather than at the individual scale. These models can be seen as the large population size limit of stochastic Markov epidemiological models [48]. Clearly, stochastic dispersal, contagion and single stochastic extinction events play a major role at very early stages of the epidemic. Our study focused on a period of time when the epidemic is already well-established in the population. Therefore, we were not aiming here at describing the beginning of the epidemic. Actually, data at the county scale are not available for this early stage.

Deterministic models are numerically more tractable and simplify parameter estimation procedures. Thus, they seem more adapted for our purpose. However, we note that according to recent studies [49], stochastic SIR models are Markovian only if the infectious periods follow independent exponential distributions, which may not be realistic. Other more general assumptions, with e.g. a bimodal distribution of the sojourn time in the infectious compartment lead to more complex macroscopic integro-differential models [49]. Nonetheless, the estimated parameter values (e.g. $\mathcal{R}_0$) and the epidemic trajectories remain very close to those obtained with the simpler (Markovian) SIR model. Yet it would be interesting to carry out a similar study to ours here with this more general framework.

As underlined by our study, sanitary policies seem to play a major role compared to other (more local) covariate effects such as average temperature. Spatially differentiated policies may therefore induce a spatial heterogeneity in the contact rates. Since June 2020 (and until October 29), France has adopted local policies. One could also consider an intermediate model between $\mathcal{M}_2$ and $\mathcal{M}_3$, with a few classes of counties, each class corresponding to a level of sanitary measures. Such an approach, that would allow some intermediate spatial heterogeneity in the contact rates, would probably improve the current model $\mathcal{M}_3$ for the recent periods in France.

Our model can be used for predicting the spatial unfolding of the epidemic. However, such a forecast holds true only as long as the transmission and interaction coefficients remain constant in the future. In practice, this does not happen over long periods as there are public policy measures that keep changing over time. To wit, Schlosser et al. [50] demonstrated a strong effect of mitigation policies on long-range interactions in Germany. That affects the coefficient $w_{j,k}$ in model $\mathcal{M}_3$. In the same spirit, Roques et al. [17] showed a reduction of the contact rate due to lockdown in France. Furthermore, people may adapt their social habits because of increased awareness of the epidemic or owing to a fatigue effect. Nonetheless, the prediction is useful for the public decision process as one can predict the time scale and spatial spreading of the epidemic, if all else remains constant, that is, in the absence of policy or social changes.

Thus, the forecasting of future disease spread would require a model that takes into account the interactions between the epidemic dynamics and the disease management strategies, at a local and global scale. A possible approach would involve a coupling between an epidemiological system and a decision-making model. In this framework, the epidemiological system is an extended version of $\mathcal{M}_3$ in which the spatial heterogeneity and the intercounty transmission depend on the local management strategies. The decision-making model could follow the line of the epi-economic approach proposed in [51].

Data accessibility. All data used in this manuscript are publicly available. French mortality data at the county scale are available at https://www.gouvernement.fr/info-coronavirus/carte-et-donnees. The data necessary to generate all of

the figures in the main text and the Matlab® scripts used to produce these figures have been deposited in the Open Science Framework repository (https://osf.io/4g8p3/?view_only=c80bfa7ae6ae42ffb087d4acef3bd8ca).

Authors' contributions. All authors conceived the models, L.R. and H.B. wrote the paper, O.B. and L.R. carried out the numerical computations, L.R. and S.S. performed the statistical analysis, V.B. developed the web app, all authors discussed the results and contributed to the final manuscript.

Competing interests. We declare we have no competing interests.

Funding. This work was funded by INRAE (MEDIA network) and EHESS.

Acknowledgements. The authors are grateful to the reviewers for their insightful, detailed and very helpful comments. We thank Jean-François Rey for assistance in developing the Web app.

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
