## [Reviewer comments · Royal Society Open Science]

Review History

RSOS-201382.R0 (Original submission)

Review form: Reviewer 1

Is the manuscript scientifically sound in its present form?

Yes

Are the interpretations and conclusions justified by the results?

Yes

Is the language acceptable?

Yes

Do you have any ethical concerns with this paper?

No

Have you any concerns about statistical analyses in this paper?

No

Recommendation?

Accept with minor revision (please list in comments)

Comments to the Author(s)

The manuscript addresses the question of the spatial spreading of COVID-19 by introducing a parsimonious epidemiological modeling approach that accounts for spatial heterogeneity. Leveraging on statistical inference the numerical analysis and model selection suggest that in the secondary phase of the epidemics, local effects are secondary to processes at the global scale.

The paper is well written and the methods are clear with some exceptions that might need clarification, see below. I recommend the paper for publication, with revision as appropriate.

1-It is becoming clear the fundamental role of asymptomatic infections in the COVID-19 transmission, see e.g. Nature volume 584, pages425–429(2020). Before introducing the parsimonious model eq.0.1, on which all proposed models are based upon, I suggest spending a few sentences on the level of approximation that this model introduces. The authors here should somehow justify parsimonious modeling that is still appropriate and that is able to capture the essence of the problem under study, by referencing exiting literature, as opposed to very unrealistic oversimplifications.

2-Related to the previous point: epidemic spreading is an inherently stochastic process. The authors should specify that all their proposed models are fully deterministic and lack this important feature that actually accounts for the early-stage evolution of real epidemics and in particular of COVID-19.

Can the authors justify this choice with respect to the specific aim of this study?

3-Minor comment: in eq.0.2 ρ is not defined elsewhere before.

4-How are the weights of eq.0.3 defined? From its definition, it seems that w_{ij} does not vanish, which is counterintuitive. Are they probabilities or rates? This needs to be clarified in the text.

5-In the existing literature on epidemics spreading metapopulation models, the travel weights are usually related to the underlying graph connectivity via a power law, see e.g. PNAS March 16, 2004 101 (11) 3747-3752, as opposed to the authors' choice of a gravity-like approach. While I do not doubt the validity of this choice in the context of riot propagation analyzed in the study cited by the authors, in the present context of general human mobility it might be not appropriate. I suggest providing additional literature where this approach is used in the present context in order to better justify this choice.

6-Fig.1 shows that all models but M_0 are a good fit for real mortality data. This quite surprising considering that there are substantial differences between them. Can the author find an explanation for this result?

7-Definition of C and ρ is missing in the text.

8-I agree with the authors that the parsimonious modeling approach is able to reproduce with good accuracy the observed data on mortality in France. I am wondering how general are these conclusions. Can this model be applied well to other countries? In addition, what is the predictive power of this approach besides fitting existing data? I suggest adding a few sentences to clarify these points in the discussion.

Review form: Reviewer 2

Is the manuscript scientifically sound in its present form?

Yes

Are the interpretations and conclusions justified by the results?

No

Is the language acceptable?

Yes

Do you have any ethical concerns with this paper?

No

Have you any concerns about statistical analyses in this paper?

No

Recommendation?

Reject

Comments to the Author(s)

The manuscript presents a 4 models; in particular, a metapopulation model for the spread of COVID-19 in France. The fits of the models to data are compared to each other and it is concluded that the metapopulation model fits the data best out of the 4.

I have two main criticisms of the manuscript.

First, the metapopulation model is not a new idea in itself -- such models are widely used, although they often include a latent compartment. One thing that might be new about this model is the assumption of power-law coupling between counties. This assumption is not backed by data to a satisfactory degree. Moreover, we know that travel patterns have changed in a non-trivial way during the pandemic (see for example the preprint by Schlosser et al. (2020)). Hence, I am sorry to say that I do not see the model itself makes the paper suitable for publication in Royal Society Open Science.

The second criticism relates to the conclusions based on the interpretation of the results. The authors want to make conclusions on what measures are effective in reducing transmission and whether the spatial distribution of deaths is likely to be due to climate or not. They also suggest tracking immunity in French counties based on the model. While I agree with many of the author's views (in particular that the spatial distribution of deaths is very unlikely to be due to variations in temperature), I am not convinced that the model is a good enough representation of reality that I would trust so far-reaching conclusions. I ask myself: Would I trust the conclusions of the paper if it had concluded that climate was likely to be the driving factor of the spatial distribution of COVID-19 deaths in France. The answer would be no. The authors do not make a convincing case that the model fits reality well enough that we can derive knowledge about the virus or base societal interventions on it.

Going forward, I think the authors could potentially write a paper focusing exclusively on models. If they could show that their metapopulation model is simpler and faster than the others on the market, and that the power-law county coupling fits reality well, that would make an interesting addition to the literature.

Review form: Reviewer 3

Is the manuscript scientifically sound in its present form?

Yes

Are the interpretations and conclusions justified by the results?

Yes

Is the language acceptable?

Yes

Do you have any ethical concerns with this paper?

No

Have you any concerns about statistical analyses in this paper?

No

Recommendation?

Accept as is

Comments to the Author(s)

The paper is well written. The scientific material and methods included in the paper are well presented and explained, in particular: the modeling part as well as the mathematical analysis part. The results of the study can be considered as an original contribution in the field of mathematical modeling of infectious diseases.

Decision letter (RSOS-201382.R0)

Dear Dr Roques

The Editors assigned to your paper RSOS-201382 "A parsimonious model for spatial transmission and heterogeneity in the COVID-19 propagation" have now received comments from reviewers and would like you to revise the paper in accordance with the reviewer comments and any comments from the Editors. Please note this decision does not guarantee eventual acceptance.

Please submit your revised manuscript and required files (see below) no later than 21 days from today's (ie 15-Oct-2020) date. Note: the ScholarOne system will 'lock' if submission of the revision is attempted 21 or more days after the deadline. If you do not think you will be able to meet this deadline please contact the editorial office immediately.

on behalf of Mark Chaplain (Subject Editor)
openscience@royalsociety.org

Associate Editor Comments to Author:

Comments to the Author:

Thank you for your patience while we sought reviewers: as we're sure you can imagine, all researchers (and thus potential referees) with interests in virology and COVID in particular are extremely pressed for time at present.

In any case, we've now received a range of views on your work. Given that Reviewer 1 has a number of requests for revision, and Reviewer 2 is very critical of your work, we are not able to accept the manuscript in its current form. Indeed, given the critiques of Reviewer 2 in particular, we're erring somewhat on the side of generosity to give you a chance to revise your manuscript in line with Reviewer 1 and Reviewer 2's comments, rather than rejecting outright. Please bear in mind that we do not generally permit multiple rounds of revision, so you should do everything possible to respond to the concerns of the reviewers, as Reviewer 1 and Reviewer 2 will be asked to assess your revised paper: if they remain unsatisfied by the manuscript, we will be hard-pressed to accept it.

Reviewer comments to Author:

Reviewer: 1

Comments to the Author(s)

The manuscript addresses the question of the spatial spreading of COVID-19 by introducing a parsimonious epidemiological modeling approach that accounts for spatial heterogeneity. Leveraging on statistical inference the numerical analysis and model selection suggest that in the secondary phase of the epidemics, local effects are secondary to processes at the global scale.

The paper is well written and the methods are clear with some exceptions that might need clarification, see below. I recommend the paper for publication, with revision as appropriate.

1-It is becoming clear the fundamental role of asymptomatic infections in the COVID-19 transmission, see e.g. Nature volume 584, pages425-429(2020). Before introducing the parsimonious model eq.0.1, on which all proposed models are based upon, I suggest spending a few sentences on the level of approximation that this model introduces. The authors here should somehow justify parsimonious modeling that is still appropriate and that is able to capture the essence of the problem under study, by referencing exiting literature, as opposed to very unrealistic oversimplifications.

2-Related to the previous point: epidemic spreading is an inherently stochastic process. The authors should specify that all their proposed models are fully deterministic and lack this important feature that actually accounts for the early-stage evolution of real epidemics and in particular of COVID-19.

Can the authors justify this choice with respect to the specific aim of this study?

3-Minor comment: in eq.0.2 rho is not defined elsewhere before.

4-How are the weights of eq.0.3 defined? From its definition, it seems that w_{ii} does not vanish, which is counterintuitive. Are they probabilities or rates? This needs to be clarified in the text.

5-In the existing literature on epidemics spreading metapopulation models, the travel weights are usually related to the underlying graph connectivity via a power law, see e.g. PNAS March 16, 2004 101 (11) 3747-3752, as opposed to the authors' choice of a gravity-like approach. While I do not doubt the validity of this choice in the context of riot propagation analyzed in the study cited by the authors, in the present context of general human mobility it might be not appropriate. I suggest providing additional literature where this approach is used in the present context in order to better justify this choice.

6-Fig.1 shows that all models but M0 are a good fit for real mortality data. This quite surprising considering that there are substantial differences between them. Can the author find an explanation for this result?

7-Definition of C and rho is missing in the text.

8-I agree with the authors that the parsimonious modeling approach is able to reproduce with good accuracy the observed data on mortality in France. I am wondering how general are these conclusions. Can this model be applied well to other countries? In addition, what is the predictive power of this approach besides fitting existing data? I suggest adding a few sentences to clarify these points in the discussion.

Reviewer: 2

Comments to the Author(s)

The manuscript presents a 4 models; in particular, a metapopulation model for the spread of COVID-19 in France. The fits of the models to data are compared to each other and it is concluded that the metapopulation model fits the data best out of the 4.

I have two main criticisms of the manuscript.

First, the metapopulation model is not a new idea in itself -- such models are widely used, although they often include a latent compartment. One thing that might be new about this model is the assumption of power-law coupling between counties. This assumption is not backed by data to a satisfactory degree. Moreover, we know that travel patterns have changed in a non-trivial way during the pandemic (see for example the preprint by Schlosser et al. (2020)). Hence, I am sorry to say that I do not see the model itself makes the paper suitable for publication in Royal Society Open Science.

The second criticism relates to the conclusions based on the interpretation of the results. The authors want to make conclusions on what measures are effective in reducing transmission and whether the spatial distribution of deaths is likely to be due to climate or not. They also suggest tracking immunity in French counties based on the model. While I agree with many of the author's views (in particular that the spatial distribution of deaths is very unlikely to be due to variations in temperature), I am not convinced that the model is a good enough representation of reality that I would trust so far-reaching conclusions. I ask myself: Would I trust the conclusions of the paper if it had concluded that climate was likely to be the driving factor of the spatial distribution of COVID-19 deaths in France. The answer would be no. The authors do not make a convincing case that the model fits reality well enough that we can derive knowledge about the virus or base societal interventions on it.

Going forward, I think the authors could potentially write a paper focusing exclusively on models. If they could show that their metapopulation model is simpler and faster than the others on the market, and that the power-law county coupling fits reality well, that would make an interesting addition to the literature.

Reviewer: 3

Comments to the Author(s)

The paper is well written. The scientific material and methods included in the paper are well presented and explained, in particular: the modeling part as well as the mathematical analysis part. The results of the study can be considered as an original contribution in the field of mathematical modeling of infectious diseases.

===PREPARING YOUR MANUSCRIPT===

===PREPARING YOUR REVISION IN SCHOLARONE===

<https://royalsociety.org/journals/authors/author-guidelines/#supplementary-material> to include a suitable title and informative caption. An example of appropriate titling and captioning may be found at https://figshare.com/articles/Table_S2_from_Is_there_a_trade-off_between_peak_performance_and_performance_breadth_across_temperatures_for_aerobic_sc_ope_in_teleost_fishes_/3843624.

Author's Response to Decision Letter for (RSOS-201382.R0)

See Appendix A.

RSOS-201382.R1 (Revision)

Review form: Reviewer 2

Is the manuscript scientifically sound in its present form?

Yes

Are the interpretations and conclusions justified by the results?

Yes

Is the language acceptable?

Yes

Do you have any ethical concerns with this paper?

No

Have you any concerns about statistical analyses in this paper?

No

Recommendation?

Accept with minor revision (please list in comments)

Comments to the Author(s)

I thank the authors for responding thoroughly to my previous comments. The replies and corrections improved my view of the work and I would like to congratulate the authors with the fine paper.

The one remaining major reservation I have with this manuscript relates to the Discussion section.

Throughout the manuscript, I think the authors do a good job of presenting their work clearly and with the necessary caveats. In the Discussion, I find that the authors conclude too boldly based on their analysis.

More concretely, the authors write

"Hence, we find that initial conditions and spatial diffusion are the main drivers of the spatial pattern of the COVID-19 epidemic."

In my opinion, this is much too strong of a statement than the study warrants. The conclusions should reflect the (very reasonable) simplifications the analysis were done under. Hence, I would suggest that the authors change this strong statement into something like,

"Our analysis of 4 simplified mathematical models suggest that initial conditions and spatial diffusion are likely to be the main drivers of the spatial pattern of the COVID-19 epidemic."

A second example of bold conclusions occur further down

"Our results indicate that — at the country scale — travel restriction alone, although they may have a significant effect on the cumulative number of deaths and the reproduction number over a definite period, are less efficient than social distancing and other sanitary measures."

This again, is a very strong conclusion that a modelling study like this is not well-suited to make. I think a more fair representation of the (interesting) results would be,

"Our results indicate that — at the country scale — travel restriction alone, although they may have a significant effect on the cumulative number of deaths and the reproduction number over a definite period, are less efficient in reducing transmission in the simple mathematical models than social distancing and other sanitary measures."

A third example is

"For instance, in the most impacted county the immunity rate is 16%, whereas it is less than 1% in less affected counties."

A more representative statement would, in my view, be

"For instance, in the most impacted county, our mathematical model's best estimate is that the immunity rate is 16%, whereas it is less than 1% in less affected counties."

In addition to these, I have the following comments.

1.

From the authors' reply to my last comments and questions, I understand that the authors are not particularly interested in the effect of climate and temperature. I think what gave me this impression is the first paragraph of the Discussion. The details about temperature variations seem like a detour. I would suggest that the authors remove the sentences

"For instance, the mean temperature during the considered period ranges from 12.0°C to 18.4°C depending on the region. We do observe a negative correlation between the mean temperature and the immunity rate (see Fig. S1), however it does not reflect a causality."

I would also encourage the authors to come up with one more example of local covariates and include this in the sentence as follows

"Actually, our study shows that if there is an effect of local covariates such as the mean temperature __or ... __ on the spread of the disease once it emerged, its effect is of lower importance compared to the global processes at work at the country scale, such as sanitary measures"

I think this might avoid the confusion.

2.

I find the following sentence imprecise:

"Herd immunity requires that a fraction $1 - 1/R_0 \approx 70\%$ of the population has been infected."

While this is true for simple well-mixed models, the threshold could be different when network effects and other heterogeneities are included (see for example (Britton, Ball and Trapman, 2020)). Once again, I would suggest a more cautious statement such as,

"Back-of-the-envelope estimates suggest that herd immunity requires that a fraction of approximately $1 - 1/R_0 \approx 70\%$ of the population has been infected."

3.

Lastly, the authors write

"Remarkably, the estimated levels of immunity are comparable to those observed in Spain by population-based serosurveys [45], with values ranging from 0.5% to 13.0% at the beginning of May at the provincial resolution. Such a large-scale serological testing campaign has not yet been carried out in France. However, if such data become available in France, our predictions could be evaluated and our model updated accordingly by including this new dataset in our estimation procedure."

The following study came out following the submission of the present manuscript, so I would not demand a discussion of its result. If possible, however, I think a single sentence in the Discussion commenting on the findings of Le Vu et al.'s recent preprint [www.medrxiv.org/content/10.1101/2020.10.20.20213116v1] would be great. Fig 3 in that manuscript seems encouraging at first sight.

Decision letter (RSOS-201382.R1)

Dear Dr Roques

On behalf of the Editors, we are pleased to inform you that your Manuscript RSOS-201382.R1 "A parsimonious approach for spatial transmission and heterogeneity in the COVID-19 propagation" has been accepted for publication in Royal Society Open Science subject to minor revision in accordance with the referees' reports. Please find the referees' comments along with any feedback from the Editors below my signature.

Please submit your revised manuscript and required files (see below) no later than 7 days from today's (ie 02-Dec-2020) date. Note: the ScholarOne system will 'lock' if submission of the revision is attempted 7 or more days after the deadline. If you do not think you will be able to meet this deadline please contact the editorial office immediately.

on behalf of Mark Chaplain (Subject Editor)
openscience@royalsociety.org

Reviewer comments to Author:
Reviewer: 2

Comments to the Author(s)

I thank the authors for responding thoroughly to my previous comments. The replies and corrections improved my view of the work and I would like to congratulate the authors with the fine paper.

The one remaining major reservation I have with this manuscript relates to the Discussion section.

Throughout the manuscript, I think the authors do a good job of presenting their work clearly and with the necessary caveats. In the Discussion, I find that the authors conclude too boldly based on their analysis.

More concretely, the authors write

"Hence, we find that initial conditions and spatial diffusion are the main drivers of the spatial pattern of the COVID-19 epidemic."

In my opinion, this is much too strong of a statement than the study warrants. The conclusions should reflect the (very reasonable) simplifications the analysis were done under. Hence, I would suggest that the authors change this strong statement into something like,

"Our analysis of 4 simplified mathematical models suggest that initial conditions and spatial diffusion are likely to be the main drivers of the spatial pattern of the COVID-19 epidemic."

A second example of bold conclusions occur further down

"Our results indicate that — at the country scale— travel restriction alone, although they may have a significant effect on the cumulative number of deaths and the reproduction number over a definite period, are less efficient than social distancing and other sanitary measures."

This again, is a very strong conclusion that a modelling study like this is not well-suited to make. I think a more fair representation of the (interesting) results would be,

"Our results indicate that — at the country scale — travel restriction alone, although they may have a significant effect on the cumulative number of deaths and the reproduction number over a definite period, are less efficient in reducing transmission in the simple mathematical models than social distancing and other sanitary measures."

A third example is

"For instance, in the most impacted county the immunity rate is 16%, whereas it is less than 1% in less affected counties."

A more representative statement would, in my view, be

"For instance, in the most impacted county, our mathematical model's best estimate is that the immunity rate is 16%, whereas it is less than 1% in less affected counties."

In addition to these, I have the following comments.

1.

From the authors' reply to my last comments and questions, I understand that the authors are not particularly interested in the effect of climate and temperature. I think what gave me this impression is the first paragraph of the Discussion. The details about temperature variations seem like a detour. I would suggest that the authors remove the sentences

"For instance, the mean temperature during the considered period ranges from 12.0°C to 18.4°C depending on the region. We do observe a negative correlation between the mean temperature and the immunity rate (see Fig. S1), however it does not reflect a causality."

I would also encourage the authors to come up with one more example of local covariates and include this in the sentence as follows

"Actually, our study shows that if there is an effect of local covariates such as the mean temperature __or ... __ on the spread of the disease once it emerged, its effect is of lower importance compared to the global processes at work at the country scale, such as sanitary measures"

I think this might avoid the confusion.

2.

I find the following sentence imprecise:

"Herd immunity requires that a fraction $1 - 1/R_0 \approx 70\%$ of the population has been infected."

While this is true for simple well-mixed models, the threshold could be different when network effects and other heterogeneities are included (see for example (Britton, Ball and Trapman, 2020)).

Once again, I would suggest a more cautious statement such as,

"Back-of-the-envelope estimates suggest that herd immunity requires that a fraction of approximately $1 - 1/R_0 \approx 70\%$ of the population has been infected."

3.

Lastly, the authors write

"Remarkably, the estimated levels of immunity are comparable to those observed in Spain by population-based serosurveys [45], with values ranging from 0.5% to 13.0% at the beginning of May at the provincial resolution. Such a large-scale serological testing campaign has not yet been carried out in France. However, if such data become available in France, our predictions could be evaluated and our model updated accordingly by including this new dataset in our estimation procedure."

The following study came out following the submission of the present manuscript, so I would not demand a discussion of its result. If possible, however, I think a single sentence in the Discussion commenting on the findings of Le Vu et al.'s recent preprint

[www.medrxiv.org/content/10.1101/2020.10.20.20213116v1] would be great. Fig 3 in that manuscript seems encouraging at first sight.

===PREPARING YOUR MANUSCRIPT===

one version identifying all the changes that have been made (for instance, in coloured highlight, in bold text, or tracked changes); a 'clean' version of the new manuscript that incorporates the changes made, but does not highlight them. This version will be used for typesetting.

===PREPARING YOUR REVISION IN SCHOLARONE===

Author's Response to Decision Letter for (RSOS-201382.R1)

See Appendix B.

Decision letter (RSOS-201382.R2)

Dear Dr Roques,

It is a pleasure to accept your manuscript entitled "A parsimonious approach for spatial transmission and heterogeneity in the COVID-19 propagation" in its current form for publication in Royal Society Open Science.

COVID-19 rapid publication process:

We are taking steps to expedite the publication of research relevant to the pandemic. If you wish, you can opt to have your paper published as soon as it is ready, rather than waiting for it to be published the scheduled Wednesday.

This means your paper will not be included in the weekly media round-up which the Society sends to journalists ahead of publication. However, it will still appear in the COVID-19 Publishing Collection which journalists will be directed to each week (<https://royalsocietypublishing.org/topic/special-collections/novel-coronavirus-outbreak>).

If you wish to have your paper considered for immediate publication, or to discuss further, please notify openscience_proofs@royalsociety.org and press@royalsociety.org when you respond to this email.

on behalf of Prof Mark Chaplain (Subject Editor)
openscience@royalsociety.org

Appendix A

Message to the Editor

Dear Editor,

We are grateful to the three reviewers for their comments.

1) We answered all of the points raised by reviewer 1 in the response below. Their comments were most helpful for us to improve our manuscript. In particular, we have now added references that support several of our assumptions, we clarified the parameter definitions where needed, and we added a discussion about the possible extensions of our approach to a broader context.

2) Reviewer 2 raised four important points. Namely, a) the novelty of the model, b) the justification of the power-law coupling assumption, c) whether we produce enough evidence to substantiate our conclusions, d) the potential use of our model for decision making.

We addressed all of these points in the response to the reviewer below. We just would like to emphasize some elements here. Regarding the first point raised by the referee, we agree that the epidemiological model *per se* is not new. In the originally submitted version of our manuscript, we indeed referred to the classical article of Kendall, and we also mentioned a work on riots with a similar model. What is new in our approach however is to put together several methods, namely, an extension of these models involving ordinary differential systems with continuous time-varying coefficients and spatial interaction with a reduced number of corresponding variables, a probabilistic observation model, and a statistical estimation procedure. In fact, to the best of our knowledge, even overlooking the other aspects, we have not seen a model that combines both continuous time dependent coefficients and spatial interactions. As a matter of fact, such a model is rather delicate because it may involve a host of parameters and there is a risk of overfitting from the observations. This is why it has to be combined with the other aspects of our approach. To be sure, such a model is needed when analyzing the effect in behaviour changes caused by the imposition of sanitary rules.

Additionally, the main motivation of our paper does not lie in the models themselves but in the application to COVID-19 epidemic. Comparing different models and selecting one appears to us as very useful. This new approach indeed allows us to gain knowledge about specific features of the epidemic. In particular this works shows the overwhelming effect of global processes due in particular to sanitary measures when compared to local covariates. These conclusions are supported by statistical tests. Nonetheless, we do not suggest here that our model should be taken as the sole source for policy decisions. Of course, in such a matter of uttermost importance, regarding policy and their consequences, governments should rely on an array of approaches, not just one. In this respect, bringing this approach in the game is very useful.

3) We are very happy to read the comment by Reviewer 3. It seems to indicate that we reached the goals we set to ourselves in taking up this work. We are thankful to the reviewer for it.

Looking forward to receiving your comments,
On behalf of the authors,

Responses to the reviews

Here are more detailed answers for the reviewers.

In the ms., we highlight in red all changes and additions. In our responses, we endeavor to reply to all the reviewer comments and describe the changes.

Reviewer: 1

We are thankful to the reviewer for their comments, which were most helpful for us to improve our manuscript.

The paper is well written and the methods are clear with some exceptions that might need clarification, see below. I recommend the paper for publication, with revision as appropriate.

Reviewer's comment: 1-It is becoming clear the fundamental role of asymptomatic infections in the COVID-19 transmission, see e.g. Nature volume 584, pages 425–429(2020). Before introducing the parsimonious model eq.0.1, on which all proposed models are based upon, I suggest spending a few sentences on the level of approximation that this model introduces. The authors here should somehow justify parsimonious modeling that is still appropriate and that is able to capture the essence of the problem under study, by referencing existing literature, as opposed to very unrealistic oversimplifications.

Our answer.

We added a paragraph in the Introduction to answer this point (page 2).

“It is widely accepted that the age structure of the population [11] and the presence of asymptomatic infectious individuals [12] affect the dynamics of the COVID-19 epidemic. Several works take these features into account (e.g., [13,14] for models with several age classes and [15] for models with a specific compartment of unreported cases). However, these approaches bring in a new difficulty in that they involve a larger number of unknown parameters leading to identifiability issues and underdetermination of the models by the data. As is often the case, there is a trade-off between searching for a more realistic description and models that can be trusted because they still match the data satisfactorily with a small set of parameters to identify. From this point of view, we observe that more parsimonious approaches, in which the above-mentioned compartments (e.g. reported and unreported cases) are merged into a single ‘infectious’ compartment, are still able to capture key features of the epidemic directly from surveillance data, in some cases even earlier than more complex approaches. For instance, with a simple SIRD model coupled with a probabilistic observation model, [5] estimated the infection fatality ratio (IFR) at an early stage of the epidemic, and obtained results that matched subsequent analyses based on more detailed and realistic models such as in the study of Institut Pasteur [3] that involved several age classes and a very precise description of the transition from hospital admission to ICU and death. It turned out

that in both cases, the authors obtained exactly the same IFR of 0.5% (excluding deaths in nursing homes). Yet, the simple SIRD model hinged on a reduced dataset (number of tests, number of positive cases and number of deaths in France) as compared to the study [3], which required additional data in France and also data from the Diamond Princess cruise ship. Thus, Occam's razor leads us here to give preference to the simplest model whenever it is sufficient to fit the data.”

See also our answer to the point 2 below (regarding stochastic models).

Reviewer's comment: 2-Related to the previous point: epidemic spreading is an inherently stochastic process. The authors should specify that all their proposed models are fully deterministic and lack this important feature that actually accounts for the early-stage evolution of real epidemics and in particular of COVID-19. Can the authors justify this choice with respect to the specific aim of this study?

Our answer. We added two paragraphs in the Discussion to answer this point: page 12.

“We considered here a macroscopic deterministic framework. That is, we used continuous models at the population scale rather than at the individual scale. These models can be seen as the large population size limit of stochastic Markov epidemiological models [47]. Clearly, stochastic dispersal, contagion and single stochastic extinction events play a major role at very early stages of the epidemic. Our study focused on a period of time when the epidemic is already well-established in the population. Therefore we were not aiming here at describing the beginning of the epidemic. Actually, data at the county scale are not available for this early stage.

Deterministic models are numerically more tractable and simplify parameter estimation procedures. Thus, they seem more adapted for our purpose. However, we note that according to recent studies [48], stochastic SIR models are Markovian only if the infectious periods follow independent exponential distributions, which may not be realistic. Other more general assumptions, with e.g. a bimodal distribution of the sojourn time in the infectious compartment lead to more complex macroscopic integro-differential models [48]. Nonetheless, the estimated parameter values (e.g. R_0) and the epidemic trajectories remain very close to those obtained with the simpler (Markovian) SIR model. Yet it would be interesting to carry out a similar study to ours here with this more general framework. “

Reviewer's comment: 3-Minor comment: in eq.0.2 rho is not defined elsewhere before.

Our answer.

This has been corrected (page 5).

“The parameter $\rho(t)$ is a 'local' contact rate, i.e., the contact rate corresponding to the interactions between the susceptible individuals of the county k with the infectious individuals from the same county. The weights w_{jk} are dimensionless and describe the ratios between the contact rates associated with the j - k interactions and those associated with the k - k interactions. We assume a power law decay of these weights.”

Reviewer's comment: 4-How are the weights of eq.0.3 defined? From its definition, it seems that w_{ij} does not vanish, which is counterintuitive. Are they probabilities or rates? This needs to be clarified in the text.

Our answer.

With our definition, the weights w_{jk} are dimensionless and describe the ratio between the contact rates associated with the j - k interactions and those associated with the (local) k - k interactions. This is why $w_{jj}=1$ in this approach. If all of the other weights are 0 (no intercounty transmission), the model reduces to the model M1. We added in the text (page 5) that “**The weights $w_{\{j,k\}}$ are dimensionless and describe the ratios between the contact rates associated with the j - k interactions and those associated with the (local) k - k interactions.**”

5-In the existing literature on epidemics spreading metapopulation models, the travel weights are usually related to the underlying graph connectivity via a power law, see e.g. PNAS March 16, 2004 101 (11) 3747-3752, as opposed to the authors' choice of a gravity-like approach. While I do not doubt the validity of this choice in the context of riot propagation analyzed in the study cited by the authors, in the present context of general human mobility it might be not appropriate. I suggest providing additional literature where this approach is used in the present context in order to better justify this choice.

Our answer.

We agree with the referee that the gravity-like approaches neglect some phenomena such as long-distance dispersal and antagonisms between cities. Ideally, the best approach to deal with human epidemic spread would be to estimate the weights from another independent dataset (e.g., from telecom data).

However, there is a rich literature that justifies our choice of a gravity-like model for human epidemics. We added several sentences on page 5, which contains several references as requested by the referee.

“This type of power-law decay is commonly used in 'gravity models' to describe the flow of people from one area to another [30]. Though promising extensions have been proposed, with for instance higher-order interactions between cities [31], this type of interactions remains a standard choice in infectious disease modelling (e.g., [32–34]).”

6-Fig.1 shows that all models but M0 are a good fit for real mortality data. This quite surprising considering that there are substantial differences between them. Can the author find an explanation for this result?

Our answer.

That model M2 is accurate is no wonder as it involves an exceeding number of parameters. Now, we too were rather surprised by the good accuracy of models M1 and M3. The explanation is precisely one of the findings of our paper, i.e., that global processes had much more effect than local processes. See also our answer to point 8 below.

7-Definition of C and rho is missing in the text.

Our answer.

This has been corrected (page 5 for rho) and page 7 for C.

“This constant C can be interpreted as the fraction of the overall contact rate $\alpha(t)$ which can be attributed to within-county transmissions.”

8-I agree with the authors that the parsimonious modeling approach is able to reproduce with good accuracy the observed data on mortality in France. I am wondering how general are these conclusions. Can this model be applied well to other countries? In addition, what is the predictive power of this approach besides fitting existing data? I suggest adding a few sentences to clarify these points in the discussion.

Our answer.

As we mentioned in the second paragraph of the Discussion (page 11), we believe that these conclusions also hold true for other countries with the same structure as France. Namely, we think of countries or regions with a mid-size territory (in particular to avoid too contrasted climate effects within the same country) and which have a certain unity in terms of public health measures. As underlined by our study, sanitary policies seem to play a major role compared to other (more local) covariate effects such as average temperature. Spatially-differentiated policies may therefore induce a spatial heterogeneity in the contact rates. Since June 2020 (and until October 29), France has adopted local policies. One could also consider an intermediate model ‘M2.5’ with a few classes of departments, each class corresponding to a level of sanitary measures. Such an approach, that would allow some intermediate spatial heterogeneity in the contact rates, would probably improve the current model M3 for the recent periods in France.

Regarding the second point, we believe indeed that our model can be used for predicting the spatial unfolding of the epidemic. However, such a forecast holds true only if the transmission and interaction coefficients remain constant in the future. In practice, this does not happen as there are public policy measures that keep changing over time, and also people may adapt their social habits to the presence of the epidemic. Nonetheless, the prediction is useful for the public decision process as one can predict the time scale and spatial spreading of the epidemic, *if all else remains constant*, that is, in absence of policy or social changes.

Thus, the forecasting of future disease spread would require a model that takes into account the interactions between the epidemic dynamics and the disease management strategies, at a local and global scale. A possible approach would involve a coupling between an epidemiological model (e.g. the extended version of M3 in which the spatial heterogeneity depends on the local management strategies, as discussed above) and a decision-making model, inspired by epi-economic approach proposed in <https://doi.org/10.1073/pnas.1011250108>.

We added the following paragraphs (Discussion, page 13)

“As underlined by our study, sanitary policies seem to play a major role compared to other (more local) covariate effects such as average temperature. Spatially-differentiated policies

may therefore induce a spatial heterogeneity in the contact rates. Since June 2020 (and until October 29), France has adopted local policies. One could also consider an intermediate model between M2 and M3, with a few classes of departments, each class corresponding to a level of sanitary measures. Such an approach, that would allow some intermediate spatial heterogeneity in the contact rates, would probably improve the current model M3 for the recent periods in France. “

and (Discussion, page 13)

“Our model can be used for predicting the spatial unfolding of the epidemic. However, such a forecast holds true only as long as the transmission and interaction coefficients remain constant in the future. In practice, this does not happen over long periods as there are public policy measures that keep changing over time. To wit, [49] demonstrated a strong effect of mitigation policies on long-range interactions in Germany. That affects the coefficient $w_{j,k}$. In the same spirit, [17] showed a reduction of the contact rate due to lockdown in France. Furthermore, people may adapt their social habits because of increased awareness of the epidemic or owing to a fatigue effect. Nonetheless, the prediction is useful for the public decision process as one can predict the time scale and spatial spreading of the epidemic, *if all else remains constant*, that is, in absence of policy or social changes.

Thus, the forecasting of future disease spread would require a model that takes into account the interactions between the epidemic dynamics and the disease management strategies, at a local and global scale. A possible approach would involve a coupling between an epidemiological system and a decision-making model. In this framework, the epidemiological system is an extended version of M3 in which the spatial heterogeneity and the intercounty transmission depend on the local management strategies. The decision making model could follow the line of the epi-economic approach proposed in [50].”

Reviewer: 2

The manuscript presents 4 models; in particular, a metapopulation model for the spread of COVID-19 in France. The fits of the models to data are compared to each other and it is concluded that the metapopulation model fits the data best out of the 4.

I have two main criticisms of the manuscript.

Reviewer's comment: First, the metapopulation model is not a new idea in itself -- such models are widely used, although they often include a latent compartment. One thing that might be new about this model is the assumption of power-law coupling between counties. This assumption is not backed by data to a satisfactory degree. Moreover, we know that travel patterns have changed in a non-trivial way during the pandemic (see for example the preprint by Schlosser et al. (2020)). Hence, I am sorry to say that I do not see the model itself makes the paper suitable for publication in Royal Society Open Science.

Our answer.

We agree with the referee that the epidemiological model *per se* is not new.

However, as the referee points out, and as far as we know, the model is new in that it combines both continuous time dependent epidemiological coefficients and spatial interaction with a reduced number of corresponding variables. The latter is achieved by the special form of the coupling between counties. We note that, to be sure, such a model is needed when analyzing the effect in behaviour changes caused by the imposition of sanitary rules. Yet, such a model is rather delicate to handle. Indeed, it may involve a host of parameters because of the time dependence and there is a risk of overfitting from the observations.

This is why it has to be combined with the other aspects of our approach. Indeed, what is also new in our approach is to bring together several methods and approaches. Indeed, we combine this extension of the epidemiological models, a probabilistic observation model, and a statistical estimation procedure.

Comparing different models and selecting one appears to us as very useful. A great variety of models are being used in the present covid-19 pandemic. They range from the simplest SIR model to models with a large number of compartments and detailed behavioral coefficients. See for instance the new paragraphs we have added on page 2 (Introduction) and page 12 (Discussion). Thus, it is important to quantitatively analyze to what degree the various models are precise while being coefficients thrifty. In this respect, our study can inspire future works, in particular to further compare other approaches as well.

In order to dispel any ambiguity, we made it clear in our manuscript that we do not claim the novelty of the ODE system in itself. In particular, we replaced 'model' by 'approach' in the title of our ms, we replaced 'This original spatially parsimonious model suggests' by 'This suggests' in the abstract, and we added several references to other models with spatial interactions, see page 5.

Regarding the justification of the power-law coupling assumption, there are many references that advocate their use for the spread of human epidemics. We added some relevant ones (page 5). We thank the reviewer for pointing out the reference Schlosser et al. (2020) which is quite interesting. We certainly agree that lockdowns have induced strong changes in mobility. This is precisely why we considered scenarios where the weights w_{ij} (which control intercounty transmission) can be modified. Related to this point, we added a paragraph in the Discussion (page 13) to point out the difficulty of forecasting the future spread of the epidemic owing to this temporal variability:

*“Our model can be used for predicting the spatial unfolding of the epidemic. However, such a forecast holds true only as long as the transmission and interaction coefficients remain constant in the future. In practice, this does not happen over long periods as there are public policy measures that keep changing over time. To wit, [49] demonstrated a strong effect of mitigation policies on long-range interactions in Germany. That affects the coefficient $w_{j,k}$. In the same spirit, [17] showed a reduction of the contact rate due to lockdown in France. Furthermore, people may adapt their social habits because of increased awareness of the epidemic or owing to a fatigue effect. Nonetheless, the prediction is useful for the public decision process as one can predict the time scale and spatial spreading of the epidemic, *if all else remains constant*, that is, in absence of policy or social changes.”*

Reviewer's comment: The second criticism relates to the conclusions based on the interpretation of the results. The authors want to make conclusions on what measures are effective in reducing transmission and whether the spatial distribution of deaths is likely to be due to climate or not. They also suggest tracking immunity in French counties based on the model. While I agree with many of the author's views (in particular that the spatial distribution of deaths is very unlikely to be due to variations in temperature), I am not convinced that the model is a good enough representation of reality that I would trust so far-reaching conclusions. I ask myself: Would I trust the conclusions of the paper if it had concluded that climate was likely to be the driving factor of the spatial distribution of COVID-19 deaths in France. The answer would be no. The authors do not make a convincing case that the model fits reality well enough that we can derive knowledge about the virus or base societal interventions on it.

Our answer.

Please note that we do not only address the effect of temperatures. More generally we emphasize the overwhelming effect of global (country-scale) processes due in particular to public health measures as compared to the effect of local covariates (including temperatures).

There are two points in this comment. The first one is whether we produce enough evidence to substantiate our conclusions. These rely on two statistical criteria. Both provide strong indications in favour of the models that do not take the local heterogeneity into account. An important advantage in using statistical criteria such as AIC and BIC is that they were conceived to avoid over-parametrization, and to select as it were the 'true process' that generated the data (see ref [40]). From a pragmatic viewpoint, the notion of 'true process' has to be understood as 'likely process'. Since these criteria only provide us with a model comparison, they only yield specific conclusions. They allow us to state that global processes are more likely to have shaped the spatial distribution than local covariates (M1 vs M2). Indeed, introducing a local effect does not improve the AIC and BIC values. Furthermore, intercounty transmission has played a significant role in the spread of the epidemics (M1 vs M3). In our view, this brings some new insight on the virus spreading. Now, of course, this 'knowledge' could be further challenged in further studies. We clarified these points in the Result section (page 8).

Anecdotally, we wish to report that at the outset of this work, we actually believed that it was necessary to take local processes into account to accurately render the local epidemic dynamics. In fact, we were rather surprised by the outcome of our study that showed that models with global parameters fit the data as well as a fully heterogeneous model.

The second point of the comment regards the potential use of our model for decision making. We do not suggest here that our model should be taken as the sole source for policy decisions. Nonetheless, having such a parameter thrifty model, simple to implement and not requiring large sociological studies to understand people's behavior, is immensely useful. It allows one to test at once various scenarios, and to easily conduct simulations. Of great value is the indication of spatial spreading that could allow advance actions in some regions. It should furthermore be noted that spatial diffusion is often overlooked by the very detailed behavioral models. Of course, in such a matter of uttermost importance, regarding policy and their consequences, governments should rely on an array of approaches, not just one. In this respect, bringing this approach in the game is useful. But, of course, e.g. in France, the ones provided by Institut Pasteur and INSERM are certainly essential.

Related to this point, we added a paragraph in the Discussion (page 13) arguing that forecasting the epidemic should rely on coupling the epidemiological approach and a decision-making model. Such a coupling may be used to inform decision makers for sanitary policy:

“Thus, the forecasting of future disease spread would require a model that takes into account the interactions between the epidemic dynamics and the disease management strategies, at a local and global scale. A possible approach would involve a coupling between an epidemiological system and a decision-making model. In this framework, the epidemiological system is an extended version of M3 in which the spatial heterogeneity and the intercounty transmission depend on the local management strategies. The decision making model could follow the line of the epi-economic approach proposed in [50].”

Reviewer's comment: Going forward, I think the authors could potentially write a paper focusing exclusively on models. If they could show that their metapopulation model is simpler and faster than the others on the market, and that the power-law county coupling fits reality well, that would make an interesting addition to the literature.

Our answer.

We thank the reviewer for this suggestion. Actually, this is the spirit of our study, where 4 existing models are compared and tested. We agree that such an analysis based on a broader class of models would be a nice contribution to the literature, and may help scientists in the future to choose between a wide range of modelling options. However, we believe that this goes beyond the scope of the present paper. It represents an important research direction for the future as a continuation of our paper. Nevertheless, we added several paragraphs dealing with other models, from more complex models involving age classes or a specific compartment of unreported cases (Introduction, page 2) to stochastic and non-Markovian models (Discussion, page 12). We also discussed how our approach may be improved to deal with more recent data (Discussion, page 13).

Reviewer: 3

Comments to the Author(s)

The paper is well written. The scientific material and methods included in the paper are well presented and explained, in particular: the modeling part as well as the mathematical analysis part. The results of the study can be considered as an original contribution in the field of mathematical modeling of infectious diseases.

Our answer.

We are very happy to read this comment. For us, it seems to indicate that we reached the goals we set to ourselves in taking up this work. We are thankful to the reviewer for it.

Appendix B

Message to the Editor

Dear Editor,

We are much grateful to the three reviewers for their comments. We truly appreciate their work and are quite impressed by the quality of the review process. In particular, the reaction of ref. 2 is remarkable and we truly appreciate it. All the comments are pertinent and the new wording that the ref. 2 suggests is very nice.

In the "Response to review" file please find the answers to the remarks of the referee.

In the new version, we highlight in red all changes and additions. We also added a sentence to say that we are grateful to the referees who helped us improve our ms.

Thank you very much for your great handling of our paper.

Henri Berestycki and Lionel Roques

Response to the review

Reviewer 2:

We are very grateful to the reviewer for their reaction. We feel that all suggestions and criticisms were very much to the point and they helped us improve our manuscript. All the comments on the revised version are pertinent and the new wording that the ref. 2 suggests is very nice. We have essentially followed all their formulations.

- 1. Reviewer's comment: More concretely, the authors write "Hence, we find that initial conditions and spatial diffusion are the main drivers of the spatial pattern of the COVID-19 epidemic." In my opinion, this is much too strong of a statement than the study warrants. The conclusions should reflect the (very reasonable) simplifications the analysis were done under. Hence, I would suggest that the authors change this strong statement into something like, "Our analysis of 4 simplified mathematical models suggest that initial conditions and spatial diffusion are likely to be the main drivers of the spatial pattern of the COVID-19 epidemic."*

Our answer: This observation has been taken into account (second paragraph, page 11). We have just inserted "are likely to be".

- 2. Reviewer's comment: A second example of bold conclusions occur further down "Our results indicate that — at the country scale— travel restriction alone, although they*

may have a significant effect on the cumulative number of deaths and the reproduction number over a definite period, are less efficient than social distancing and other sanitary measures." This again, is a very strong conclusion that a modelling study like this is not well-suited to make. I think a more fair representation of the (interesting) results would be, "Our results indicate that — at the country scale — travel restriction alone, although they may have a significant effect on the cumulative number of deaths and the reproduction number over a definite period, are less efficient in reducing transmission in the simple mathematical models than social distancing and other sanitary measures."

Our answer: This observation has been taken into account (third paragraph, page 12) and we have changed the sentence along this line.

- 3. Reviewer's comment: A third example is "For instance, in the most impacted county the immunity rate is 16%, whereas it is less than 1% in less affected counties." A more representative statement would, in my view, be "For instance, in the most impacted county, our mathematical model's best estimate is that the immunity rate is 16%, whereas it is less than 1% in less affected counties."*

Our answer: This observation has been taken into account (last paragraph, page 11).

- 4. Reviewer's comment: From the authors' reply to my last comments and questions, I understand that the authors are not particularly interested in the effect of climate and temperature. I think what gave me this impression is the first paragraph of the Discussion. The details about temperature variations seem like a detour. I would suggest that the authors remove the sentences "For instance, the mean temperature during the considered period ranges from 12.0°C to 18.4°C depending on the region. We do observe a negative correlation between the mean temperature and the immunity rate (see Fig. S1), however it does not reflect a causality." I would also encourage the authors to come up with one more example of local covariates and include this in the sentence as follows "Actually, our study shows that if there is an effect of local covariates such as the mean temperature ___ or ... ___ on the spread of the disease once it emerged, its effect is of lower importance compared to the global processes at work at the country scale, such as sanitary measures"*

Our answer: This observation has been taken into account: we removed the first sentence with the details on the mean temperature value, but we kept the second one to refer to Figure S1, and, as suggested, we added another example of covariate, namely the age structure (first paragraph, page 11).

- 5. Reviewer's comment: I find the following sentence imprecise: "Herd immunity requires that a fraction $1 - 1/R_0 \approx 70\%$ of the population has been infected." While this is true for simple well-mixed models, the threshold could be different when network effects and other heterogeneities are included (see for example (Britton, Ball*

and Trapman, 2020)). Once again, I would suggest a more cautious statement such as, "Back-of-the-envelope estimates suggest that herd immunity requires that a fraction of approximately $1 - 1/R_0 \approx 70\%$ of the population has been infected."

Our answer: We adopted this formulation (last paragraph, page 11).

6. *Reviewer's comment: Lastly, the authors write "Remarkably, the estimated levels of immunity are comparable to those observed in Spain by population-based serosurveys [45], with values ranging from 0.5% to 13.0% at the beginning of May at the provincial resolution. Such a large-scale serological testing campaign has not yet been carried out in France. However, if such data become available in France, our predictions could be evaluated and our model updated accordingly by including this new dataset in our estimation procedure." The following study came out following the submission of the present manuscript, so I would not demand a discussion of its result. If possible, however, I think a single sentence in the Discussion commenting on the findings of Le Vu et al.'s recent preprint [www.medrxiv.org/content/10.1101/2020.10.20.20213116v1] would be great. Fig 3 in that manuscript seems encouraging at first sight.*

Our answer: We thank the reviewer for pointing out this interesting reference which is indeed very relevant. We added the following sentences (second paragraph, page 12):
"Actually, a very recent study [Le Vu et al, 2020] (October 2020) provides estimates for prevalence of anti-SARS-CoV-2 antibodies in France. It shows that the nationwide seroprevalence was 4.93% ([4.02-5.89]) by mid-May. This is consistent with the immunity rate given by our model at the same dates 4.51%. The distribution of the immunity rate in Fig. 3 here also seems in good qualitative agreement with the findings of this study."